# Piezoelectric Micromachined Ultrasonic Transducer-Integrated Helmholtz Resonator with Microliter-Sized Volume-Tunable Cavity

**DOI:** 10.3390/s22197471

**Published:** 2022-10-02

**Authors:** Guo-Hua Feng, Wen-Sheng Chen

**Affiliations:** 1Department of Power Mechanical Engineering, Institute of Nano Engineering and MicroSystems, National Tsing Hua University, Hsinchu 30013, Taiwan; 2Department of Mechanical Engineering, National Chung Cheng University, Chiayi 621301, Taiwan

**Keywords:** Helmholtz resonator, piezoelectric film, two-term power series curve, ultrasonic, volume-tunable cavity

## Abstract

In this study, a piezoelectric micromachined ultrasonic transducer (PMUT) is integrated with a microliter-sized volume-tunable Helmholtz resonator. The passive Helmholtz resonator is constructed using an SU8 photolithography-defined square opening plate as the neck portion, a 3D-printed hollow structure with a threaded insert nut, and a precision set screw to form the volume-controllable cavity of the Helmholtz resonator. The fabricated piezoelectric films acted as ultrasonic actuators attached to the surface of the neck SU8 plate. Experimental results show that the sound pressure level (SPL) and operation bandwidth could be effectively tuned, and a 200% SPL increase and twofold bandwidth enhancement are achieved when setting the cavity length to 0.75 mm compared with the open-cavity case. A modified Helmholtz resonator model is proposed to explain the experimental results. The adjusting factors of the effective mass and viscous damper are created to modify the existing parameters in the conventional Helmholtz resonator model. The relationship between the adjusting factors and cavity length can be described well using a two-term power series curve. This modified Helmholtz resonator model not only provides insight into this active-type Helmholtz resonator operation but also provides a useful estimation for its optimal design and fabrication.

## 1. Introduction

The Helmholtz resonator is an air-filled cavity with an opening. When an exterior sound wave pressurizes the air around the narrow portion of the cavity (called the neck), the air in this neck portion oscillates and causes an air pressure increase inside the cavity while the resonant condition is matched [1,2,3].

Conventional Helmholtz resonators are typically used for sound absorption [4,5,6,7]. For example, Zhao and Morgans [8] used Helmholtz resonators as passive dampers. The frequency of the resonator with maximum damping was also altered by varying the area of the Helmholtz resonator neck. Tang et al. [9] found that separating the cavity into compartments could improve the sound absorption capacity and expand the working bandwidth of a Helmholtz resonator.

Recently, numerous sensors have been deployed in IoT energy harvesting. Powering these extensively distributed devices has become increasingly important. Acoustic energy harvesting along with the Helmholtz resonator has become an interesting research topic [10,11,12,13,14,15]. For instance, Matova et al. [16] reported energy harvesting from airflow using a micromachined piezoelectric harvester inside a Helmholtz resonator. This airflow angle toward the entrance of the fluid flow into the neck of the Helmholtz resonators affected the sensitivity of energy harvesting. Cui et al. [17] demonstrated an acoustic energy harvester based on the integration of an acoustic metasurface and Helmholtz resonator. The acoustic metasurface converges acoustic waves at the center of the metasurface, and the Helmholtz resonator further enhances the energy-harvesting efficiency.

Some approaches to assembling the ultr2asonic transducer with the Helmholtz resonator structure and enhancing the performance of ultrasonic emission have been based on the actuator being attached to the cavity portion of the Helmholtz resonator. Although this configuration could increase the operation bandwidth compared with using the resonator cavity without a neck portion, the reduced pressure magnitude may be a shortcoming [18].

Micromachining technology allows cantilever elements to be integrated with the neck portion of the Helmholtz resonator with relatively little complication. With this configuration, a piezoresistive cantilever integrated with a micro Helmholtz resonator was investigated for use as an ultrasonic proximity sensor [19]. Driving a piezoelectric cantilever above the neck of a Helmholtz resonator has also been demonstrated [20]. A larger displacement of the piezoelectric cantilever can be achieved, but the operation of acoustic waves at the planned frequency of the Helmholtz resonator with optimal performance required precise control of the fabrication parameters.

A four-beam connected circular piezoelectric micromachined ultrasonic transducer (PMUT) on a titanium foil integrated with a cavity-volume controllable Helmholtz resonator was proposed [21]. The SU8-made hollow plate formed the neck portion of the Helmholtz resonator. The stereolithography 3D-printed base was assembled with the threaded insert nut, and hexagon socket headless set screws were structured as the volume-tunable cavity of the Helmholtz resonator (Figure 1). Experimental data show that the developed device not only increases the sound pressure but also broadens the operation bandwidth. Most importantly, we proposed a modified Helmholtz resonator model to bridge the gap between the experimental data and the calculated results based on the conventional Helmholtz resonator model. The proposed model could provide insight into a micro-sized ultrasonic actuator integrated with a Helmholtz resonator with a microliter-sized volume cavity.

This paper is the extended version of our previous IEEE conference paper [21]. In this study, we have added more simulation and experimental data as well as proposed a mathematical model (Section 4) to describe and analyze the experimental results. In Section 4, through mathematical modeling, we gain insight into the relationship between the geometrical parameters of the PMUT-Helmholtz resonator design and operating frequency, damping ratio, as well as sound pressure. This could benefit the optimal device design of the micro-sized ultrasonic actuator integrated with a microliter-sized volume cavity Helmholtz resonator to obtain better performance.

The application of the developed devices would be applied to a biomimetic sonar system for the navigation of robots. By tuning the peak frequencies of individual devices, we could manufacture an array of ultrasonic emitters with a broad frequency band and a designable sound level distribution. Moreover, for the device used as an air sonar emitter, the fine-tuning of peak frequency allows the user to design the carrier frequency with a larger range. This gives more choices for the range of Doppler frequency shifts in practical air sonar applications.

## 2. Device Design and Fabrication

A 4 µm thick titanium foil was used as the substrate. The PMUT element was fabricated on this foil using the following design: the dimensions of each element were first simulated by finite element analysis (COMSOL) to achieve a fundamental frequency above 20 kHz [22]. The square element had a side length of 1.7 mm. Each element had a circle in the middle with a radius of 0.8 mm. Four beams with a width of 0.8 mm were connected to the circle. Four alignment marks were made around each PMUT for subsequent assembly with a Helmholtz resonator.

For the COMSOL simulation, the studied composite film for PMUT vibration included four layers, silver, PZT, titanium, and PZT (Figure 2a). The boundary condition was set fixed at four edges of the four-beam structure (Figure 2b). We utilized the COMSOL default algorithm to generate mesh on the structure. The material parameters were set as below. The densities of silver, titanium, and PZT were 4506, 10,500, and 7750 kg/m^3^, respectively. The silver and titanium films were considered isotropic with Young’s moduli of 83 GPa and 117.5 GPa and Poisson ratios of 0.37 and 0.32, respectively. The stiffness matrix (Cij) of the PZT film was C11 = C22 = 120.3 GPa, C12 = 75.2 GPa, C13 = C23 = 72 GPa, C33 = 115 GPa, C44 = C55 = 22.2 GPa, C66 = 33.4 GPa. Figure 2c shows the simulation results of the mode shapes of the first, second, third, and fourth resonance. The corresponding resonant frequencies were 22.336, 44.214, 44.215, and 55.450 kHz, respectively (Figure 2c).

We also performed the simulation of the PMUT mounted on an open cavity with different cavity lengths. The simulation results show the resonant frequency changed while the cavity length altered. The fundamental frequencies for the cases of the open-cavity lengths of 10 mm, 5 mm, 3 mm, and 1.5 mm were 23.904 kHz, 23.919 kHz, 28.914 kHz, and 28.873 kHz (Figure 2d).

A micromachining process was used to fabricate the patterned foil substrates (Figure 3). The commercially available 4 μm thick titanium foil was used and we cut it to a proper size to be anchored on a glass plate with four pieces of tape attaching the edge parts of the titanium foil to the glass plate for better handling. After the standard photolithography process, wet etching was performed. The patterned titanium foil was removed from the glass plate and rinsed with deionized (DI) water for piezoelectric film deposition [23].

We employed a hydrothermal method to grow a piezoelectric film on a titanium substrate [24,25]. A precursor solution was then prepared. The content of the precursor solution used for growing PZT film involved four chemicals, zirconium oxychloride, lead nitrate, titanium dioxide, and potassium hydroxide. A total of 26.14 g of zirconium oxychloride and 41.91 g of lead nitrate were mixed with 250 mL of DI water by stirring for 10 min. Then, 1.45 g of TiO_2_ powder was added to the resultant solution and continuously stirred for 40 min. Subsequently, 103.29 g of potassium hydroxide was mixed with 200 mL of DI water and added to the previously processed solution. After stirring for 50 min, the processed titanium substrate array was placed into an autoclave to hydrothermally grow the PZT film for 40 h at 180 °C.

Subsequently, a shadow mask was fabricated for electrode deposition. It was designed to allow 200 μm between the electrode and the edge of the designed PMUT element. A 1 μm thick silver film was then sputtered onto the PZT film as the top electrode. At this point, the PMUT elements on the thin foil were ready.

A hollow plate was fabricated as the neck portion of the Helmholtz resonator. The spin-coated 250 μm thick SU8 film was spin-coated on a metal sheet, and photolithography was utilized to form hollow plates. The individual SU8 hollow plates were released from the metal sheet via wet etching.

To create the volume-tunable cavity of the Helmholtz resonator, we plotted a hollow cuboid base using Solidwork software and fabricated it using a stereolithography 3D printing machine. The hollow part was a cylinder with an opening diameter of 3 mm. An M3-sized threaded insert nut was fitted into the hollow portion of the cuboid. Then, a hexagonal socket headless set screw was inserted. The length of the hollow portion of the cuboid could be accurately controlled by adjusting the rotation angle of the screw. This allowed a volume-tunable cavity of the Helmholtz resonator to be obtained for subsequent experiments.

Finally, two thin epoxy layers were applied on both sides of the SU8 plate and attached to a diced foil containing a single PMUT element and the volume-tunable cavity of the Helmholtz resonator. The finished PMUT integrated with a volume-tunable cavity Helmholtz resonator was ready for testing. Figure 4 shows the fabricated results.

The thickness of the hydrothermally grown PZT film was approximately 7 μm as checked by a scanning electron microscope. Figure 5 shows the XRD patterns of the fabricated hydrothermal PZT films. The characterization was performed on a PANalytical X’Pert Pro XRD system for a 2θ range from 5° to 90° with a scanning speed of 4°/min and an increment of 0.02°. CuKα radiation was employed (λ = 1.5406 nm). The sharp peaks of lead zirconium titanium oxide tetragonal structure Pb(Zr_0.4_Ti_0.6_)O_3_ matching JCPDS no. 70-7730 were observed. Diffraction peaks (100), (101)/(110), (002), and (112) clearly showed in the crystalline Pb(Zr_0.4_Ti_0.6_)O_3_ thin films. The sharp peaks indicated the formation of larger grains with increased crystallinity. The XRD results indicated that the crystalline films exhibit a perovskite structure with a preferential orientation (101).

In this study, the manual assembly of the device majorly included the SU8 plate glued to the 3D-printed Helmholtz resonator, the PMUT film attached to the SU8 plate with the 3D-printed resonator, and the screw nut filling into the 3D-printed Helmholtz resonator. To mass manufacture the device, we could use robot operation and the assistance of an imaging artificial intelligence (AI) system. For example, after applying the epoxy to the surface of 3D-printed Helmholtz resonator cavity by screen-printing method, the AI-imaging system aligned the SU8 plate to the 3D-printed resonator and the robot applied the proper pressure to anchor the SU8 plate to the 3D-printed resonator. A similar scheme could be employed in the assembly of the PMUT film to the SU8 plate glued with the 3D-printed resonator.

## 3. Device Characterization

The fabricated device was characterized by simultaneously measuring the displacements of the PMUT and the generated sound pressures when driven by different electrical signals. During the measurement, different cavity volumes were adjusted. Meanwhile, the installation of the microphone, PMUT, and laser displacement meter was controlled at the same positions for each cavity-volume adjustment. The experimental setup and results are presented below.

### 3.1. Experimental Setup for Device Characterization

A laser displacement meter (model: LK-H0200, Keyence Co., Osaka, Japan) was used to obtain the displacement of the PMUT. It was installed on an XZ precision stage. The stage is produced by Newport Corp. (Irvine, CA, USA), and its motion can be controlled using a computer with a piezoelectric actuation mechanism. The travel resolution was 20 nm and the maximum stroke was 3 cm. We controlled the z-direction of the stage so that the laser spot was focused at the center of the circular foil of the PMUT for measurement (Figure 6).

A commercial free-field microphone (model: 378C01, manufacturer: PCB Piezotronics, Inc., Depew, NY, USA) was used for the sound pressure measurement. The receiving frequency spectrum is from 4 Hz to 100 kHz. The output signal from the microphone was connected to an ICP sensor signal conditioner for signal amplification. A 3D-printed holder was created to allow the microphone to anchor on the sidewall of an XZ precision stage with the required orientation and without interfering with the laser beam emitted from the laser displacement meter.

In addition, a 3D-printed rectangular hollow column was built as the test stand for placing the fabricated device. Sound absorption cotton was inserted at the bottom portion of the rectangular hollow column to avoid the effect of the emitted ultrasonic wave reflection during the open-cavity test.

The signals of both the displacement and microphone were acquired using an NI 6351 DAQ card with a sampling rate of 392 kHz. Subsequently, the recorded data were processed using MATLAB to analyze the frequency response.

### 3.2. Results of Displacement Measurement

First, a displacement experiment was performed to determine the resonances of the fabricated PMUT under the open-cavity condition (without using the set screw). The input was first set as a sinusoidal wave, with a frequency ranging from 5 to 100 kHz, and an amplitude of 12 V peak to peak (sweep: 1 s; return: 0 s; interval: 1 ms setting on the function generator Textronix AFG3022, Tektronix Inc., Beaverton, OR, USA). After the peak frequency in a certain frequency spectrum was determined, the swept frequency range was set to 5 kHz with the peak frequency being approximately centered at the swept 5 kHz.

Figure 7a shows the displacement frequency spectrum for five cases of varied cavity lengths controlled by a revolution angle of the set screw and open-cavity case. The results were directly calculated from the signal acquired by FFT without further signal processing. The fundamental resonance at 23.7 kHz can be clearly observed for the open-cavity case. We filtered the time-domain signal with a bandpass filter (−3 dB cut-off frequency at 23.5 kHz and 22.9 kHz), and a displacement of 3.3 μm at the center of the PMUT circular plate was measured (Figure 7b).

### 3.3. Result of Sound Pressure Measurement

The measured output from the microphone was a voltage signal. Figure 8a shows the frequency spectrum of the acquired signal. The open cavity means there is no enclosing structure present for the PMUT. The front side of the PMUT was in the free field. The backside of the PMUT was anchored on a structure with an opening that was much larger than the vibrational membrane of the PMUT and could be considered a free field. The backside structure is needed so that we could secure the PMUT for testing. The results that a smaller volume of the Helmholtz cavity had a larger peak frequency. All the investigated cases with the Helmholtz cavity exhibited better sound pressure levels than those without it. When adjusting the cavity length to 0.75 mm, the maximum magnitude at resonance attained an increase of 200% compared with the case of the open cavity. The corresponding amplitude of the measured voltage was 1.11 V at a frequency of 24.339 kHz. In addition, the bandwidth increases by 2.

To convert the acquired voltage signal to the decibel sound level, we divided the gain of the ICP sensor signal conditioners and used a microphone sensitivity of 2.11 (Pa/mV) provided by the manufacturer. The derived sound pressure value was then divided by the reference sound pressure of 20 μPa to calculate the decibel sound level. Figure 8b shows the sound pressure level variation and 3 dB bandwidth for different cavity volumes.

## 4. Discussion

### 4.1. Equivalent Circuit for Modeling the Developed Device

The developed device was modeled using an equivalent circuit to better understand its characteristics. The electric equivalent model can be divided into three parts for investigation: electric, mechanical, and acoustic fields (Figure 9). Electric and mechanical field modeling can be found in the literature [26,27,28]. Acoustic field modeling was first proposed. Between the electric and mechanical domains, the input voltage Vin was a sinusoidal signal set by the function generator and the imported current on the device was I. V1 represents the effective voltage that converted the force through the piezoelectric effect of the PZT film. The conversion ratio was modeled as an ideal transformer with a turn ratio ϕ, that is, *Fp* = ϕ × V_1_. The electric domain was considered as the device operation under fundamental resonance conditions. The force *Fp* caused a vibration of the PZT membrane with a velocity of u. Meanwhile, the resulting air pressure against the vibrated PZT membrane was modeled as the force *Fa*. *Fa* is the force to actuate the surrounding air at the top and bottom surfaces of the PZT membrane and is affected by environmental impedance. *Fp* due to the piezoelectric effect converted from V_1_. *Fp*-*Fa* means that the force piezoelectric effect generated minus the force actuating the surrounding air, which caused the motion of the piezoelectric membrane itself and modeled as a C_m_, R_m_, and L_m_ equivalent circuit.

*P*1 represents the air pressure on the top surface of the PZT membrane caused by the PZT membrane-produced force *Fa*. In the equivalent circuit for the acoustic field, *P*1 was the input pressure to the Helmholtz resonator. The neck portion of the Helmholtz resonator was modeled as the effect of the resistance (*R_e_*) consuming energy and the inductor (*L_e_*) conserving kinetic energy. The cavity is modeled as a capacitor (*C_e_*). We used the vibrational area of the PZT membrane as the conversion ratio, that is, *Fa* = Av × *P*1. *P*2 is pressure inside the Helmholtz cavity. *P*3 is the output pressure close to the opening region of the PMUT membrane.

If we converted the mechanical ports to acoustic ports with radiation impedance, we should consider the different surrounding conditions on the top and bottom of the four-beam PMUT membrane once the Helmholtz cavity was added. The four openings of the PMUT vibrational membrane induced the interaction of the air between the top and bottom surroundings of the membrane. This made the traditional three-port piezoelectric model with one electrical port and two mechanical ports more complicated. Using two-port representation to describe a system similar to our investigation can be found in [20,28,29,30].

We will discuss the evaluation of the elements labeled in the electric, mechanical, and acoustic-domain equivalent circuits in detail below.

Electric part:

In the electric domain, C_0_ represents the intrinsic capacitance of the PZT film, and R_0_ represents the dielectric loss in the PZT film. Because C_0_ and R_0_ are frequency dependent, we estimated these two values at the condition in which the frequency was close to the fundamental resonance and the vibration of the PZT membrane approached zero, that is u ≅ 0. Under this condition, the driving current from the input can be modeled by simply passing it through R_0_ and C_0_. In this study, we determined R_0_ and C_0_ by impedance measurement at a frequency that was approximately three times the bandwidth of the fundamental resonance below the peak frequency of the fundamental resonance.

Figure 10 shows the measured electrical impedance for the open-cavity case. The resonant frequency was 23.7 kHz, and the bandwidth was approximately 300 Hz. Thus, we selected the impedance as 22.8 kHz, which has a magnitude of 4098 Ω and an angle of −78.62°, to derive R_0_ and C_0_. The resulting R_0_ was 808.6 Ω, and C_0_ was 1.752 nF.

Mechanical Part:

Considering the PZT membrane vibration as a lumped mass–damper–spring system, the displacement of the mass was set as the displacement at the center of the membrane. When the membrane vibrates, the equation of motion is formulated as follows:(1)$$M\frac{d^{2}x\left(t\right)}{dt^{2}}+B\frac{dx\left(t\right)}{dt}+K_{s}x\left(t\right)=fp\left(t\right)-fa\left(t\right)=\Delta f\left(t\right)$$
where *M* is the mass, *B* is the viscous damper, and *Ks* is the spring constant. We can obtain the transfer function of the displacement *X*(*s*) vs. the force difference Δ*F*(*s*), which is the force difference provided by the driving force *fp* due to piezoelectric conversion from the applied driving voltage V_1_ and *fa*. This overcomes the force *fa* due to the surrounding air exerted on the vibrating membrane, as follows:(2)$$\frac{X\left(s\right)}{\Delta F\left(s\right)}=\frac{1/M}{s^{2}+\left(B/M\right)s+(K_{s}/M)}$$

Because the membrane performs a harmonic motion, its velocity and associated force can be expressed as *u(t)* = ue*^jωt^* = *jω*xe*^jωt^* and *f(t)* = Fe*^jωt^*. Thus, we have
(3)$$j\omega Mu+Bu+\frac{K_{s}}{j\omega }u=Fp-Fa$$

This expression allows us to be analogical to the inductor L_m_, resistor R_m_, and capacitor C_m_ elements in the equivalent circuit with L_m_ = M, R_m_ = B, and C_m_ = 1/Ks (Figure 9).

Meanwhile, the measured displacement of the vibrating membrane around the first resonant frequency had a relatively good fit to a second-order system. Because the actual magnitude of the measured displacement was proportional to the magnitude of the fast Fourier transform (FFT) result, we utilized parameter *K*_2_ to represent the ratio:(4)$$X\left(s\right)=K_{2}\left(X_{fft}\left(s\right)\right)=K_{2}G\left(s\right)=K_{2}\left(\frac{K_{1}\cdot \omega_{n}^{2}}{s^{2}+2\zeta \omega_{n}+\omega_{n}^{2}}\right)$$
where *ζ* is the damping ratio, *ω_n_* is the natural frequency, and *K*_1_ is a constant gain. By adjusting the *ζ*, ω_n_, and *K*_1_ values, we created a frequency response of a second-order system *G*(*s*) to match the FFT resulting curve converted from the measured displacement.

We normalized the amplitude of the FFT curves shown in Figure 7 with the near-zero slope region below the resonance (i.e., 21–22 kHz) to approximately 1 by dividing by 0.0025. This facilitates determining the damping ratio, natural frequency, and suitable gain of a corresponding second-order system to match this normalized FFT curve. Figure 8a shows that the transfer function *s*) matches the normalized curve of the open-cavity case *X_fft_(s)*. Moreover, using the relation ∆*F*(*s*)/*M* = *K*_1_*K*_2_*ω_n_*^2^, we can further derive
(5)$$K_{2}=\frac{\Delta F\left(s\right)}{K_{1}M\omega_{n}^{2}}$$

At resonance, the displacement can be obtained by measurement,
(6)$$X_{measured@resonance}=K_{2}X_{fft@resonance}=\frac{\Delta F_{@resonance}}{K_{1}M\omega_{n}^{2}}X_{fft@resonance}$$

Hence, the force applied to the membrane at resonance can be expressed as
(7)$$\Delta F_{@resonance}=K_{1}M\omega_{n}^{2}\frac{X_{measured@resonance}}{X_{fft@resonance}}$$

Because the mass of the vibrating membrane can be determined by the dimension of the membrane and the densities of the composed materials, we used *M* = 316.6 × 10^−9^ kg as the mass of the vibrating membrane for the following analysis. Thus, the ∆*F* could be obtained from the given mass *M* and ω_n_ (=2π × 23,701 rad/s), measured displacement at resonance 1.65 μm, *X_fft@resonance_* = 9.32, and for the case of open cavity: ∆*F* (*s* = jω_p_) = *Fp*(*s*)−*Fa*(*s*) = 1.714 × 10^−4^ N.

For the Helmholtz resonating cavities of varying volumes, we can find the individual *G(s)* to fit the corresponding FFT results of the measured displacement responses (Figure 11b–f). The parameters *K*_1_, *ω_n_*, ω_p_, and *ζ* of *G*(*s*) are listed in Table 1. Based on these parameters and the measured displacement at resonance, we can compute the force difference Δ*F*(*s*) for cases with different cavity lengths using the above equation.

Moreover, *Fa(s)* could be estimated as follows: according to [31], if we consider the investigated vibrating membrane of diameter a and surface vibration velocity of *v*_0_ as a small piston performing repeated back-and-forth motion and the generated sound wave as possessing the relation of the wave number, *k*, multiplying the diameter of membrane satisfies *ka* <<1, and the pressure along the axis normal to the surface of the vibrating membrane at the antinode can be expressed as
(8)$$p_{axis}=\frac{2\rho_{0}cv_{0}}{\pi a^{2}}\sin\left[\frac{k}{2}\left(\sqrt{h^{2}+a^{2}}-h\right)\right]$$
where *h* represents the distance from the membrane surface to the observation point, *v*_0_ is the vibration velocity of the membrane, c is the speed of sound, and *ρ*_0_ is the density of surrounding air. If the observation point is in the near field, *h* << *a*, and satisfies the condition *ka* < 1, the pressure can be written in the following form:(9)$$p_{axis,VNF}=\frac{\rho_{0}cv_{0}}{\pi }\frac{k}{a}$$

By dividing Equation (8) by (9), we can obtain the ratio of the pressure
(10)$$\frac{p_{axis}}{p_{axis,VNF}}=\frac{\sqrt{h^{2}+a^{2}}-h}{a}=\sqrt{{\left(\frac{h}{a}\right)}^{2}+1}-\frac{h}{a}$$

Because the measured pressure data were 15 mm above the antinode, the pressure on the antinode can then be evaluated as
*p_axis_*/*p_axis_*_,*VNF*_ = 0.0565 = −24.96 dB

Thus, we can obtain Fa according to the derived pressure at the antinode of the membrane
$$F_{a}=P_{antinode}\cdot A_{eff,\;membrane}=(P_{measured}/0.0565)\cdot A_{eff,\;membrane}\;=1.324Pa/0.0565\cdot 2.286\times 10^{-6}m^{2}=5.357\times 10^{-5}N$$

Moreover, the ratio of 0.0565 is applied to the following studied cases of the Helmholtz cavity with varied volumes to describe the pressure attenuation from the point of measured pressure to a point very close to the vibrational membrane.

The radiation impedance of the fluid surrounding the PMUT top surface was considered using Equation (10). The microphone was placed 15 mm away from the surface of the PMUT top surface. The pressure at the position very close to the PMUT surface is about 24.96 dB of the measured sound pressure (at microphone position) (*p_axis_*/*p_axis_*_,*VNF*_ = 0.0565 = −24.96 dB). In this study, the condition of *h* = 15 mm and *a* = 1.5 mm was used, some error could be induced by the assumption *h* << *a*. However, using the condition as a first-order approximation would allow us to analyze the experimental data.

Acoustic part:

A conventional Helmholtz resonator consists of neck and cavity portions. The neck of the resonator is an airflow passage that causes friction loss in the airflow. Meanwhile, the air around the neck can be considered as a finite mass with inertia that possesses kinetic energy. Therefore, the neck portion can be modeled as the effect of the resistance consuming energy and the inductor conserving kinetic energy in the equivalent circuit. The air inside the cavity of the resonator could be treated with compressible and possesses potential energy. Hence, the cavity is modeled as a capacitor in the electrical circuit. The equivalent circuit to describe the dynamic response of a Helmholtz resonator is then a resistor, inductor, and capacitor connected in series with input pressure as the input voltage and the output pressure as the voltage across the capacitor (Figure 12).

The values of the modeled RLC elements can be found in many studies. In this study, we utilized the traditional formulas below [32]:(11)$$C_{t}=\frac{V}{\rho_{air}c^{2}}$$
(12)$$L_{t}=\frac{4\rho_{air}L}{3S}$$
(13)$$R_{t}=\frac{8\pi \mu L}{S^{2}}$$

The resulting values for *C_t_*, *L_t_*, and *R_t_* are m^3^/Pa, kg/m^4^, and kg/(m^4^·s), respectively. The corresponding equivalent circuit could be expressed as shown in Figure 12.

This model described that the output sound pressure of the Helmholtz resonator can be amplified from the input sound pressure through an equivalent RLC circuit modeling and the gain can be expressed as a second-order system.

If we substituted the physical parameters listed in Table 2 using different *C_t_*, which is a function of volume change by adjusting the length change of the Helmholtz cavity, a constant *M_t_* and a constant *R_t_*, the results of the transfer function *G_t_*(*s*) = *P*2(*s*)/*P*1(*s*) are displayed in Figure 13. The results indicate that the large volume change ratio caused the peak frequency to change, ranging from 20.1 to 34.9 kHz and the gain value ranged from 779 to 450.

Examining the frequency response based on the measured output sound pressure, which was a gain of 24.96 dB multiplied by *P*3(*s*), from the studied Helmholtz resonator integrated PMUT, an approximate second-order system could be observed (Figure 8a). To further study this response, let us consider the frequency response *P*3(*s*) as a function of *P*2(*s*), or more specifically, the relation between *P*3(*s*) and *P*2(*s*) follows $P3\left(s\right)=K_{p}\left(s\right)\cdot P2\left(s\right)$, where $K_{p}\left(s\right)$ is a gain factor as a function of frequency. Thus, $P3\left(s\right)=K_{p}\left(s\right)\cdot G_{s}\left(s\right)\cdot P1\left(s\right)$, $G_{s}\left(s\right)=\frac{\omega_{n}^{2}}{s^{2}+2\zeta \omega_{n}s+\omega_{n}^{2}}$, and *P*1(*s*) was the input sound pressure.

Since the FFT result of acquired output pressure *P*3_mea_, *P*3*_fft_*, was proportional to *P*3, we tried to find a suitable second-order system to fit *P*3*_fft_* first. The ultrasonic wave signals for different cavity lengths in this investigation were analyzed. We tried to fit the frequency responses *P*3*_fft_* of measured ultrasonic wave signals with proper damping ratio *ζ* and natural frequency *ω_n_*. For all the fitting results, *P*3*_fft_* displayed a standard second-order system multiplied with a factor. Hence, we could express the fitting result of *P*3*_fft_*(*s*) as a function of *Kp*(*s*)·*Gs*(*s*)·*P*1*_fft_*(*s*). Regarding the parameter *Kp*(*s*), we could model this factor when the pressure *P*2 in the cavity transmits to the outside environment with the pressure *P*3. Because the relation between *P*3 and *P*3_fft_(*s*) can be employed a factor kf, i.e., *P*3(s) = *P*3_fft_(*s*)·kf and *P*3_fft_(*s*) = *Kp*(*s*)·*Gs*(*s*)·*P*1_fft_(*s*), under the same signal processing data points, we could use the same factor kf to relate *P*1(*s*) and *P*1*_fft_*(*s*) to better describe the relationship between *P*3 and *P*1, same as *P*3 and *P*2. In this study, kf can be evaluated as 3.912 Pa. This allowed us to estimate *P*1 over the investigated frequency range with *P*3 divided by *Gs*(*s*) and *Kp*(*s*). Figure 14 shows the results of *P*3*_fft_*(*s*) and *P*1*_fft_*(*s*) (=*P*3*_fft_*(*s*)/(*Kp*(*s*)·*Gs*(*s*))) in the frequency range of interest. Therefore, the pressure *P*2(*s*) can be described as *Gs*(*s*)·*P*1(*s*). The ratio of *P*2 to *P*1 then satisfied a standard second-order system with *P*2 as the capacitor voltage output, which was described in the traditional Helmholtz resonator model. Table 3 lists important parameters of *G*(*s*) and *Kp*.

According to the above analysis, the experimental frequency responses display a gain factor *Kp*(*s*) multiplying *Gs*(*s*), which means a gain factor should be added besides the conventional Helmholtz resonator circuit model including R, L, and C elements. To fully describe the experimental response with a proper electrical circuit model, we utilized an ideal amplifier to represent the gain factor for the modified equivalent circuit (Figure 15).

When we further examined the peak frequencies calculated based on the conventional model derived from *Gt*(*s*), the ratio of *P*2 to *P*1, though some of the peak frequency values were close to the results of *Gs*(*s*), a noticeable deviation exists for most of the studied cases. Moreover, the computed damping ratios were quite different from the *P*2 obtained by experimental results. To better describe these differences, we compared the damping ratio and resonant frequency of the RLC circuit modeled second-order system with input pressure *P*1 and output pressure *P*2 modeled as the voltage across the capacitor based on conventional theory and experimental results (Table 4).

Because the obvious deviation existed to use the conventional model to describe our developed PMUT with the integration of the Helmholtz resonator, in this study we proposed the modified values of the equivalent circuit elements of the Helmholtz resonator to replace the element values obtained in the conventional equivalent circuit to better fit the experimental results (Figure 15). In this modified circuit element, we kept the capacitance as the conventional modeling and the viscous resistance and effective mass were modified as Re and Le, respectively, which is explained in more detail below.

We proposed the concept of natural resonant frequency and damping ratio adjusting factors. This facilitated us to convert the *R_t_* and *L_t_* expressed in the conventional circuit model of the Helmholtz resonator to the modified *R_e_* and *L_e_* parameters for the experimental data fitted modified circuit model. Define the natural resonant frequency and damping ratio adjusting factors *α* and *β* as follows:(14)$$\alpha =\omega_{n,\;e}/\omega_{n,t}$$
(15)$$\beta =\zeta_{\;e}/\zeta_{t}$$
where $\omega_{n,\;e}$ and $\omega_{n,t}$ represent the natural resonant frequency derived from the experimental result and calculated from the theoretical model, respectively. $\zeta_{\;e}$ and $\zeta_{t}$ stand for the damping ratio from the experimental result and theoretical model, respectively. Moreover, if we set the capacitance *C* as a controlled variable, this means the parameters defined in the capacitance of the theoretical model were identical to the modified circuit model. Hence, the modified mass (inductor) term can be expressed as
(16)$$L_{e}=\frac{1}{\alpha^{2}\omega_{n,t}^{2}C}=\frac{1}{\alpha^{2}}L_{t}=\gamma L_{t}$$

Similarly, the modified viscous damper (resistance) term can be expressed as
(17)$$R_{e}=2\zeta_{e}\omega_{n,e}M_{e}=\frac{\beta }{\alpha }R_{t}=\delta R_{t}$$
where *γ* and *δ* are defined as the adjusting factors of the effective mass and viscous damper, respectively. Thus, we can determine the relationship between the effective mass of the Helmholtz resonator based on the experimental results and theoretical model as a function of α and the relationship between the effective viscous damper based on the experimental results and theoretical model as a function of *α* and *β*.

Table 5 lists the results of adjusting factors of natural resonant frequency, damping ratio, effective mass, and viscous damper for different volumes of the Helmholtz resonant cavity. The corresponding effective mass and viscous damper after considering the adjusting factors to fit the experimental results were also calculated. The modified effective mass and viscous damper were compared with the conventional effective mass and viscous damper. The latter parameters were much lower than the former parameters. This implied that the investigated microliter-sized Helmholtz resonator had stronger effects on viscous damping and larger effective mass compared to the regular milliliter or even greater-sized Helmholtz cavity.

To further explore the trend of the four investigated adjusting factors as a function of the volume change of the Helmholtz resonating cavity, we tried to perform the mathematical analysis by using different categories of curve fittings. A very good curve fitting result can be obtained by using a two-term power series model, $y={\mathrm{ax}}^{b}+c$ for these four adjusting factors. Figure 13 shows all fitting curves, exhibiting an R-squared value greater than 0.99. Using these fitting curve equations, we can map the conventional lumped model parameters to a new set of model parameters, allowing us to properly explain the empirical outcomes of the pressure *P*2 and *P*1 for our designed PMUT with a microliter-sized resonating resonator.

To correlate the output pressure *P*3 and cavity pressure *P*2, a circuit model of the ideal amplifier was connected to the capacitor of the acoustic model. Since the input impedance of the ideal amplifier approaches infinity, the loading effect will not result in the voltage drop across the capacitor for this modeling. This described the pressure *P*2, which was amplified through the Helmholtz cavity, converted to the output pressure *P*3 simply via a gain factor *Kp*, to accurately match the experimental results.

Experimental data also revealed a gain factor *Kp* variation accompanied by a volume change of the Helmholtz cavity. Interestingly, the relation also well fitted the two-term power series curve, $y={\mathrm{ax}}^{b}+c$ (Figure 16).

### 4.2. Validity of the Proposed Model

For the proposed model, the mechanical port and acoustic port are not independent of each other. We verify the proposed model as follows.

As described in the manuscript, *P*3 is the output pressure of the Helmholtz resonator. According to the experimental results, we have found that the output pressures of the Helmholtz resonator can be changed by adjusting the lengths (related to *C_e_*) of the designed Helmholtz resonator.

Let us use the model and mathematics below (please refer to Figure 9). Different *P*3 related to *P*1 by varied *C_e_*, *L_e_*, *R_e_*, and *Kp*. That is, *P*1 could be written as the function P1(*P*3; *C_e_*, *L_e_*, *R_e_*, *Kp*). According to the relation *Fa* = Av × *P*1 (Av is a constant), *Fa* is the function of *P*1. Using Equation (1) and considering *Fp* as a constant, as well as Equation (2), we obtain that R_m_(=B), L_m_(=*M*), and C_m_(=1/*K*_s_) are the function of X (measured displacement) and Δ*F* (function of *Fp* and *Fa*).

Thus, the mechanical port and acoustic port correlated with each other based on the parameters of L_m_, C_m_, R_m_, *L_e_*, *C_e_*, *R_e_*, and *Kp* can be verified (please check the data in Table 1 and Table 5 for more detailed numbers). Moreover, different values of L_m_, R_m_, and C_m_ obtain different *P*3 for a fixed *P*1.

Moreover, the PMUT membrane velocity u is independent of the Helmholtz design for a fixed *Fa* (please check Equations (1) and (2)). For a fixed *Fa*, (say, a fixed Δ*F* when *Fp* is considered as a constant), the membrane displacement X is a function of L_m_, C_m_, and R_m_, which have different values for different cavity lengths of the Helmholtz resonator. The magnitude of the membrane velocity u can be considered as ωX. Thus, the membrane velocity u is not independent of the Helmholtz design for a fixed *Fa*.

### 4.3. Estimation of Maximum Sound Pressure Output

According to the proposed micro Helmholtz resonator circuit model, we could evaluate the developed device to generate the maximum acoustic pressure inside the Helmholtz cavity and output pressure through the opening of the fabricated piezoelectric membrane.

Figure 17 shows the calculated results as a function of cavity length within the investigated range. As for the sound pressure from the Helmholtz resonator transmitted to the environment, the frequency responses displayed the peak magnitude increased and then decreased while the cavity length changed from 0.5 mm to 1.5 mm with an increment of 50 μm (Figure 17a). The response with a wider bandwidth for a smaller cavity volume was also monitored with the same trend as the experimental results. The largest gain occurred at 0.82 mm, which implied the gain could be even increased for the practical test (Figure 17b).

To examine the results of the sound pressure transmitted from the Helmholtz cavity to the outside environment based on the proposed modified model and experiments, Table 6 shows the comparison with error analysis. The peak frequency shows a very good matching between the model and experimental results with the largest error of less than 0.3%. The average peak gain error was 4.44% and the maximum error was kept within 10%.

Regarding the sound pressure inside the Helmholtz cavity, *P*2, we also plotted the frequency response curves based on our proposed micro Helmholtz resonator model with an increment of cavity length of 50 μm starting from the position of 0.5 mm and ending at the position of 1.5 mm (Figure 17c). The resonant frequency decreases while the cavity volume increases. This confirms the fundamental principle of Helmholtz resonance. An approximately 500 Hz peak frequency variation was observed within a 1 mm length change of the Helmholtz cavity. The cavity length greatly affected the amplified ratio of the sound pressure inside the Helmholtz cavity, especially for the smaller volume of the Helmholtz cavity (Figure 17d). The gain drops about 30 when the cavity length decreases from 0.9 mm to 0.5 mm.

## 5. Conclusions

The innovative PMUT with a volume-controllable Helmholtz resonator was developed. The displacement and sound pressure levels of the PMUT were experimentally tested. The emitted ultrasound possesses superior SPL and operational bandwidth at a properly controlled volume of Helmholtz resonator.

Based on the measured frequency responses of the piezoelectric membrane displacement and output ultrasonic pressure, we proposed a micro Helmholtz resonator model to well characterize the experimental results. This model adjusted the values of effective mass and viscous damper described in the conventional Helmholtz resonator model. The two-term power series fitting curve well maps the conventional Helmholtz resonator model parameters to our proposed micro Helmholtz resonator model. Using the proposed model, the peak frequency and magnitude could be predicted. This could be beneficial to the study of ultrasound-operated Helmholtz resonators with the actuator integrated around the neck portion of the Helmholtz resonator.

## Figures and Tables

**Figure 1 sensors-22-07471-f001:**
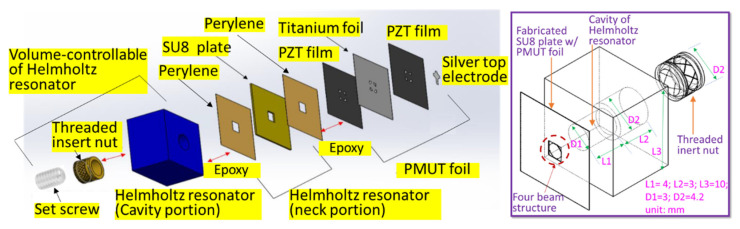
The configuration of the developed PMUT with a volume-controllable Helmholtz resonator.

**Figure 2 sensors-22-07471-f002:**
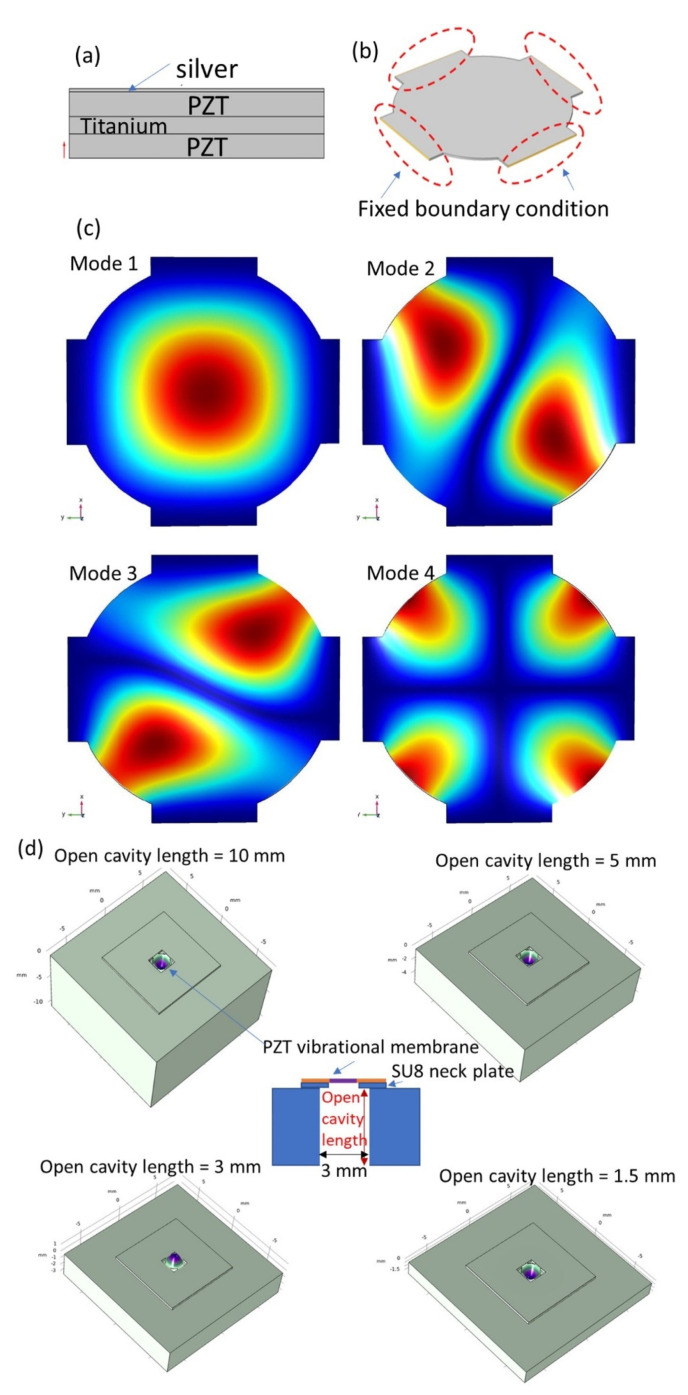
The results of COMSOL simulation. (**a**) Four layers of the PMUT film. (**b**) Boundary condition setting. (**c**) The mode shapes of the 1st, 2nd, 3rd, and 4th resonances. (**d**) The open-cavity cases with different cavity lengths.

**Figure 3 sensors-22-07471-f003:**
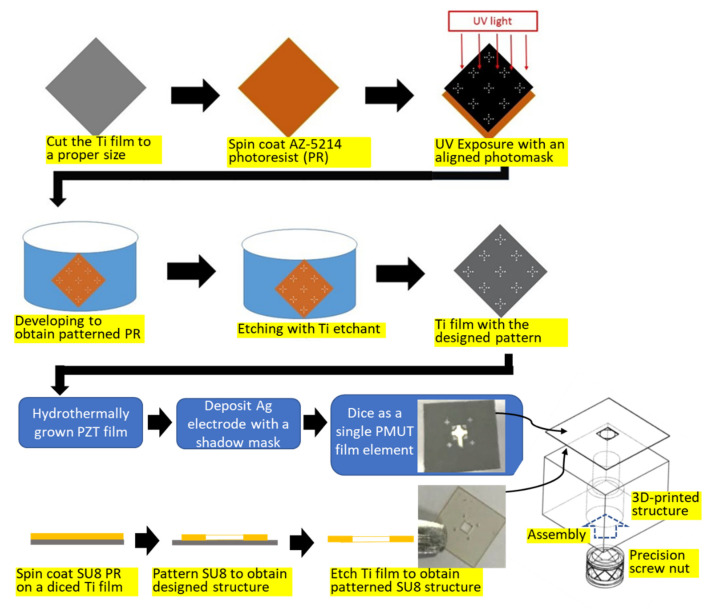
Illustration of the fabricated process for the proposed PMUT with a Helmholtz resonator.

**Figure 4 sensors-22-07471-f004:**
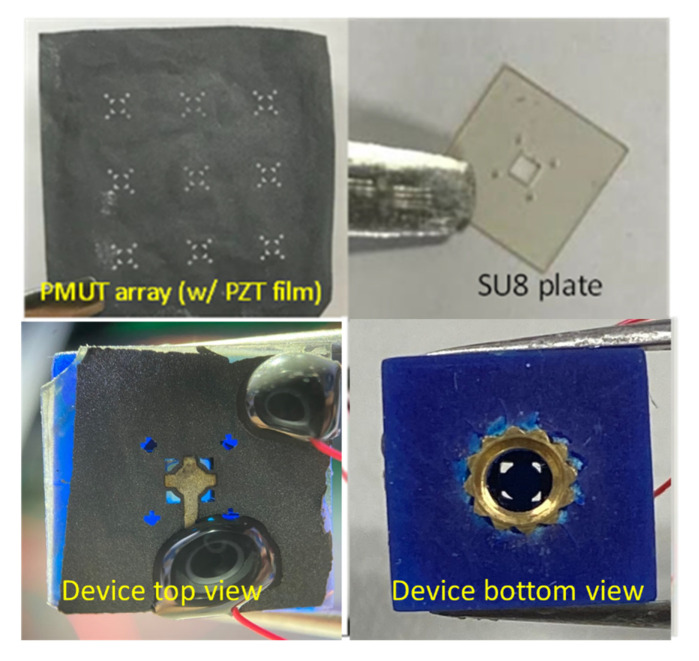
Photographs of the fabricated components and complete device.

**Figure 5 sensors-22-07471-f005:**
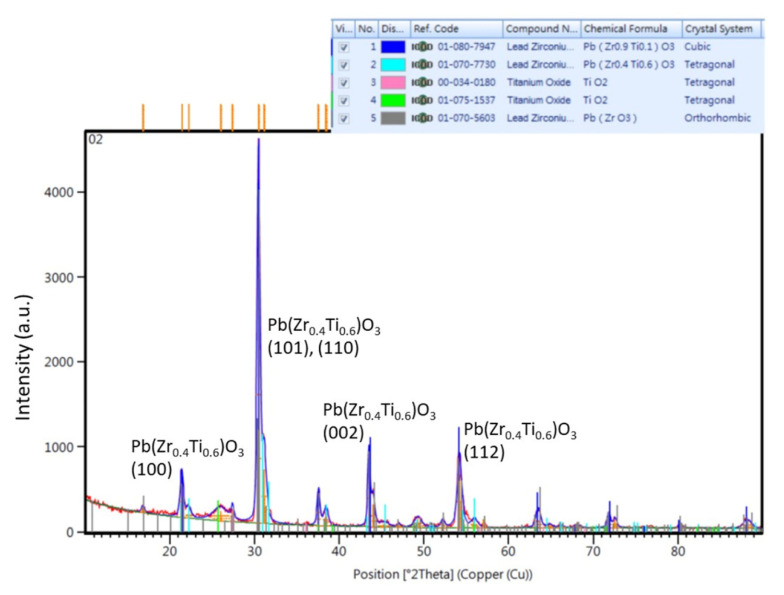
The XRD results of the fabricated PZT film.

**Figure 6 sensors-22-07471-f006:**
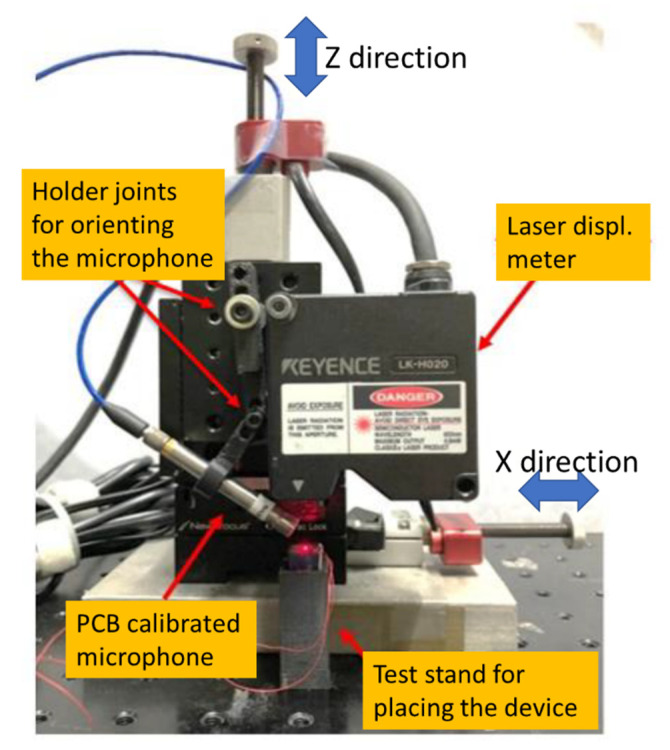
Experimental setup for acquiring the displacement and ultrasonic signals of the fabricated device.

**Figure 7 sensors-22-07471-f007:**
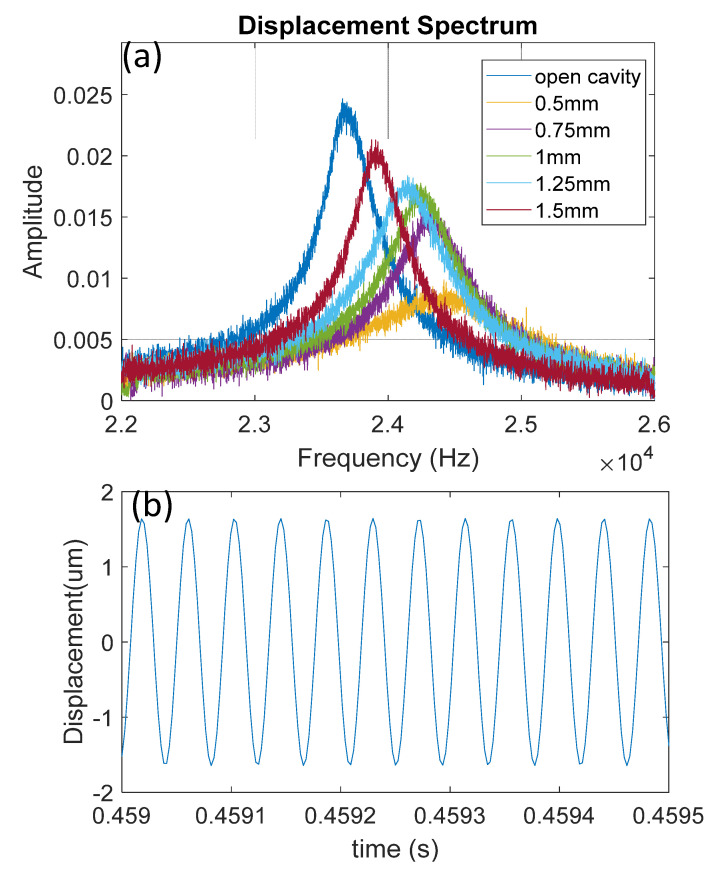
Experimental results of (**a**) displacement frequency spectrum for the varied cavity lengths of the PMUT; (**b**) measured displacement of the device at the open-cavity condition.

**Figure 8 sensors-22-07471-f008:**
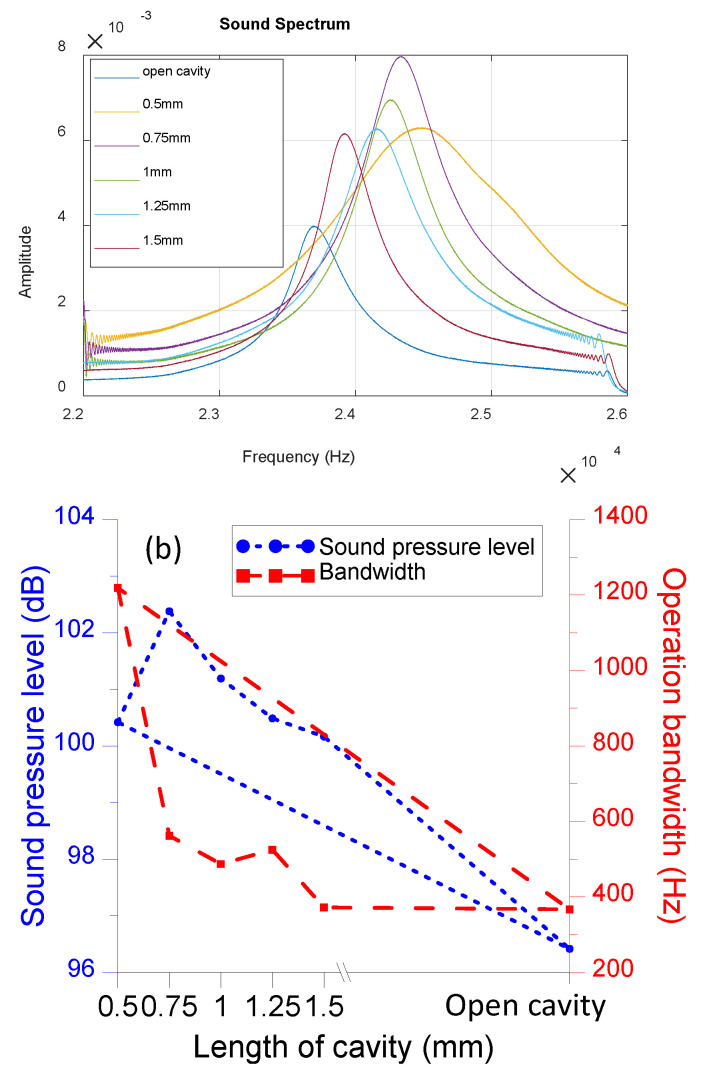
(**a**) Experimental results of sound frequency spectrum for the varied cavity lengths of the PMUT. (**b**) The sound pressure level and 3 dB bandwidth for the results in (**a**).

**Figure 9 sensors-22-07471-f009:**
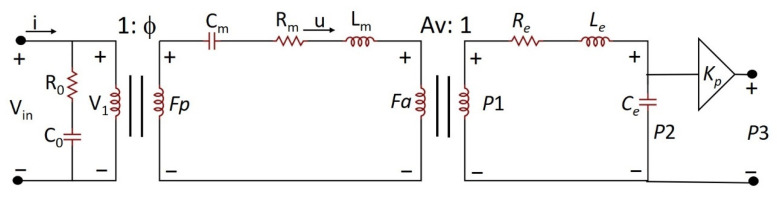
The proposed electric equivalent model of the developed PMUT integrated microliter-sized volume-tunable Helmholtz resonator.

**Figure 10 sensors-22-07471-f010:**
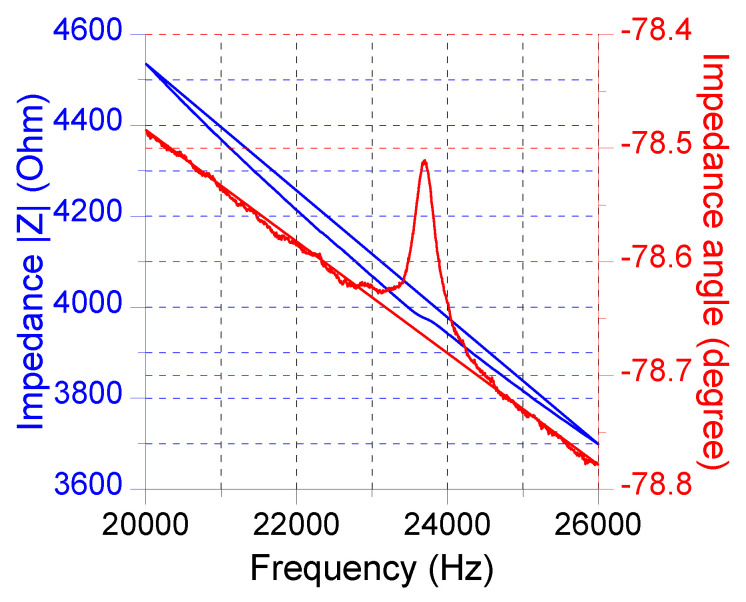
Measured electrical impedance of the PMUT device for the open-cavity case.

**Figure 11 sensors-22-07471-f011:**
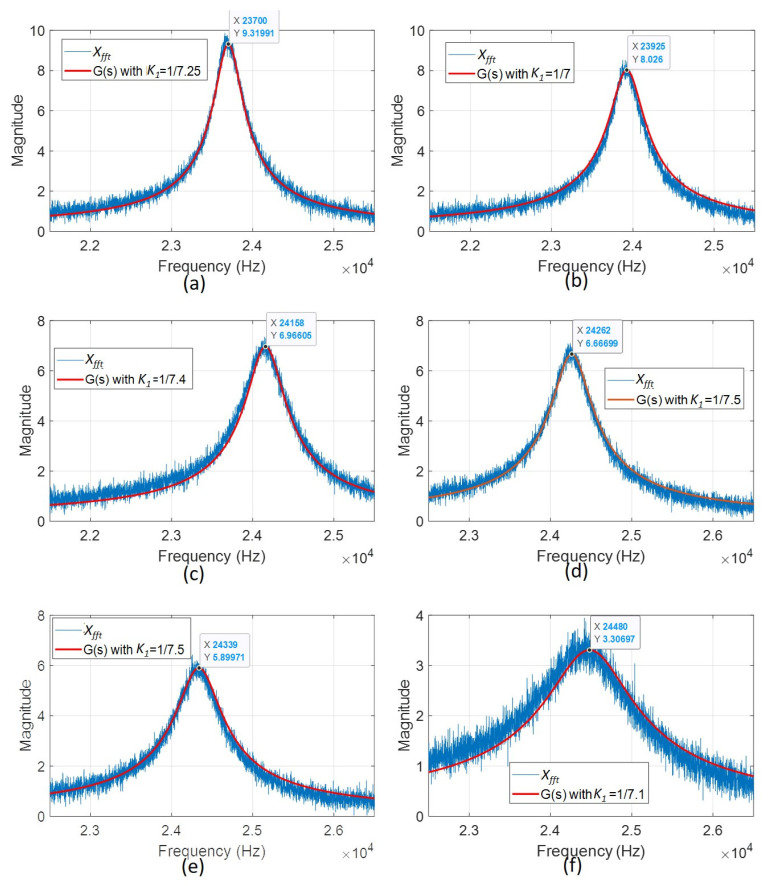
(**a**) The normalized frequency responses from *X_fft_* experimental results and their corresponding matching transfer functions *G(s)*: (**a**) the open-cavity case; (**b**) 1.5 mm; (**c**) 1.25 mm; (**d**) 1 mm; (**e**) 0.75 mm; (**f**) 0.5 mm cavity lengths.

**Figure 12 sensors-22-07471-f012:**
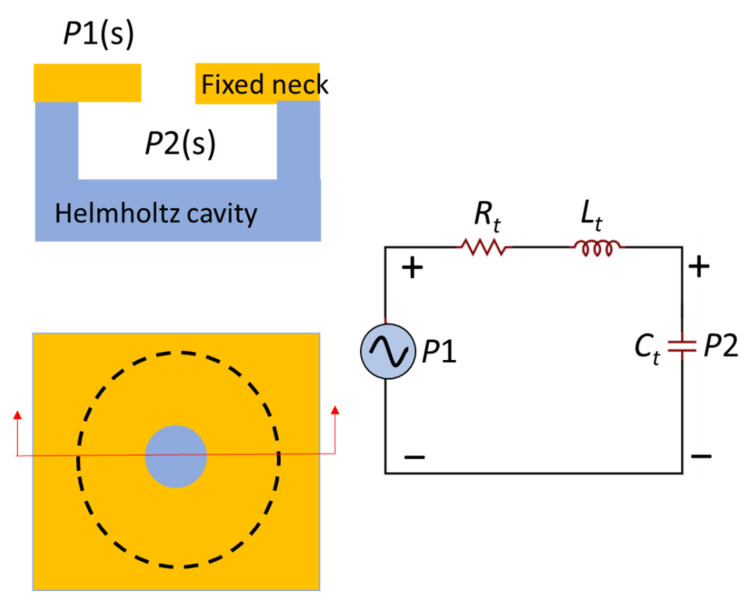
The equivalent circuit of the conventional Helmholtz resonator model.

**Figure 13 sensors-22-07471-f013:**
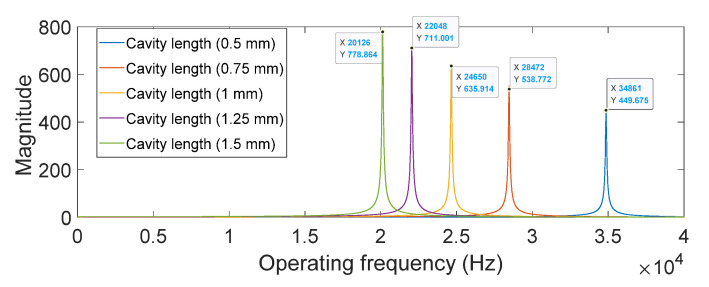
The calculated results of the transfer function *Gt*(*s*) = *P*2(*s*)/*P*1(*s*) based on the conventional Helmholtz resonator model and the physical parameters of the fabricated device.

**Figure 14 sensors-22-07471-f014:**
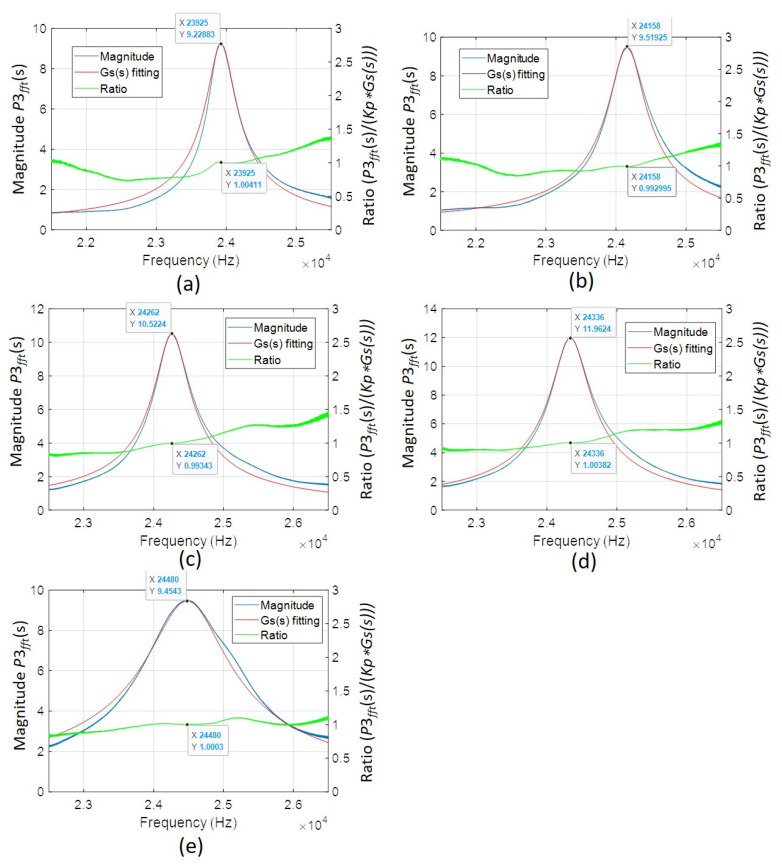
Relation between the normalized frequency responses *P*3*_fft_*(*s*) of measured sound pressures and fitting results of *Kp*(*s*) *Gs*(*s*) as well as the corresponding *P*1*_fft_*(*s*) for varied cavity lengths: (**a**) 1.5 mm; (**b**) 1.25 mm; (**c**) 1 mm; (**d**) 0.75 mm; (**e**) 0.5 mm.

**Figure 15 sensors-22-07471-f015:**
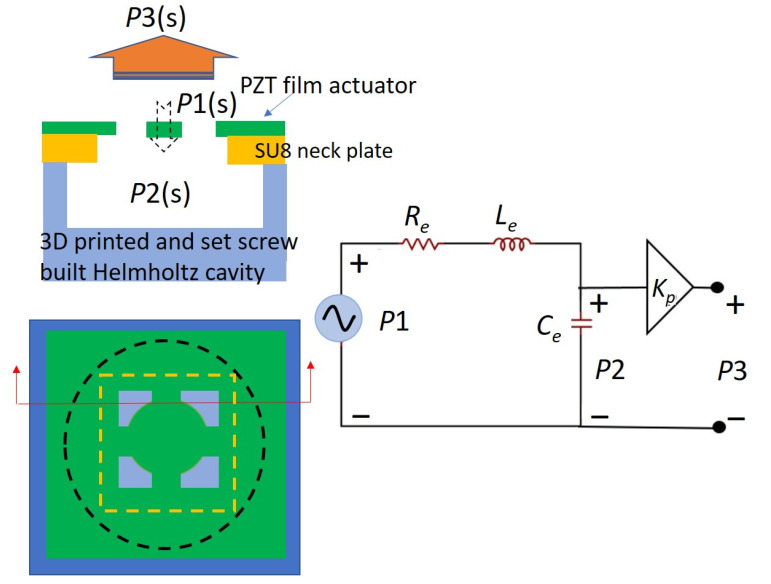
The proposed modified Helmholtz resonator circuit to characterize our experimental results.

**Figure 16 sensors-22-07471-f016:**
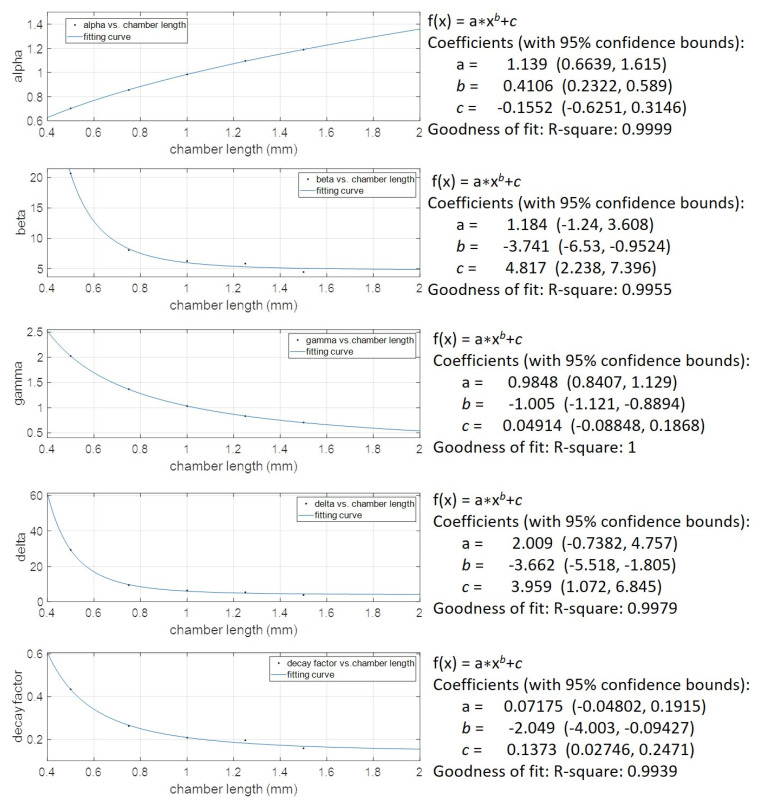
The defined adjusting factors of natural resonant frequency, damping ratio, effective mass, viscous damper, and (*P*3/*P*2) gain factor are well-fitted two-term power series curves.

**Figure 17 sensors-22-07471-f017:**
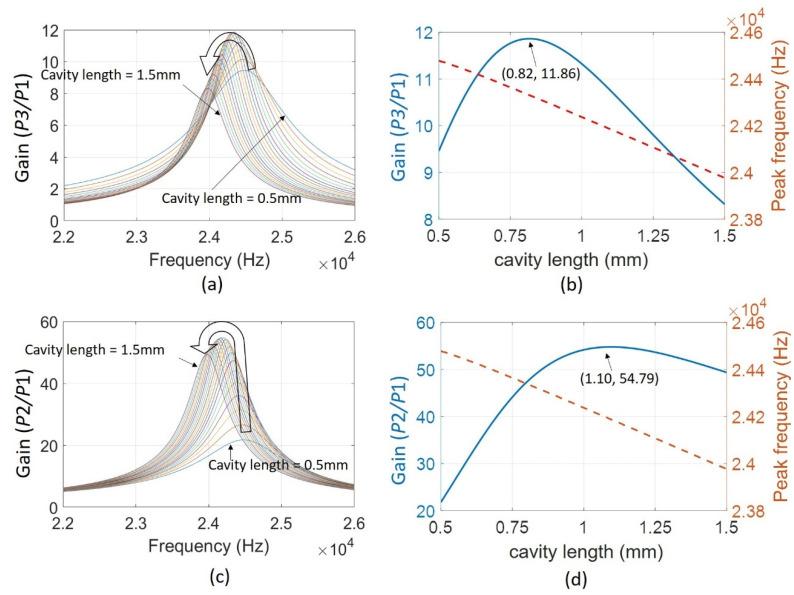
Calculated results of sound pressure gain and peak frequencies as a function of cavity length based on the modified Helmholtz resonator model for (**a**,**b**) device emitted to the surrounding; (**c**,**d**) inside the Helmholtz cavity.

**Table 1 sensors-22-07471-t001:** The parameters of *G*(*s*) to fit the studied Helmholtz resonators with varied cavity volumes.

	Damping Ratio	Natural Frequency (rad/s)	Gain *K*_1_	Resonant Frequency (rad/s)	Force Difference (N)
Open cavity	0.0074	2π × 23,701	1/7.25	2π × 23,700	1.718 × 10^−4^
1.5 mm	0.0089	2π × 23,927	1/7	2π × 23,925	1.773 × 10^−4^
1.25 mm	0.0097	2π × 24,160	1/7.4	2π × 24,158	1.889 × 10^−4^
1 mm	0.01	2π × 24,264	1/7.5	2π × 24,262	1.794 × 10^−4^
0.75 mm	0.0113	2π × 24,339	1/7.5	2π × 24,336	1.711 × 10^−4^
0.5 mm	0.0213	2π × 24,491	1/7.1	2π × 24,480	1.810 × 10^−4^

**Table 2 sensors-22-07471-t002:** The physical parameters used in the conventional Helmholtz resonator model for the studied cases.

Air density	1.184 kg/m^3^	
Sound speed	346 m/s	
Length of the neck	0.25 mm	
Cross-sectional area of the neck	2.89 mm^2^	
Effective cross-sectional area of the neck	2.89 − 2.286 = 0.604 mm^2^	
Dynamic viscosity of the air	1.849 × 10^−5^ kg/(m-s)	
Effective mass of the air in the neck, *L_t_*	836.4 kg/m^4^	
Viscous damping, R_t_	13,903 kg/(m^4^-s)	
Effective compliance of the cavity, *C_t_*, with varied lengths of cavity Lc	Lc = 0.5 mm	2.492 × 10^−14^ m^3^/Pa
	Lc = 0.75 mm	3.738 × 10^−14^ m^3^/Pa
	Lc = 1 mm	4.984 × 10^−14^ m^3^/Pa
	Lc = 1.25 mm	6.230 × 10^−14^ m^3^/Pa
	Lc = 1.5 mm	7.477 × 10^−14^ m^3^/Pa

**Table 3 sensors-22-07471-t003:** The parameters *Kp*, *ζ*, ωn, and ωp of the investigated PMUT operated in the open-cavity condition and with Helmholtz chamber for five different volumes (cavity lengths).

	Damping Ratio	Natural Frequency (rad/s)	Factor *Kp*	Resonant Frequency (rad/s)
Open chamber	0.0076	2π × 23,701	1/11	2π × 23,700
1.5 mm	0.0086	2π × 23,927	1/6.3	2π × 23,925
1.25 mm	0.0103	2π × 24,161	1/5.1	2π × 24,158
1 mm	0.0099	2π × 24,264	1/4.8	2π × 24,262
0.75 mm	0.011	2π × 24,339	1/3.8	2π × 24,336
0.5 mm	0.023	2π × 24,493	1/2.3	2π × 24,480

**Table 4 sensors-22-07471-t004:** Results of adjusting the factors of natural resonant frequency, damping ratio, effective mass, and viscous damper for varied volumes of Helmholtz resonant cavity.

Cavity Length	Conventional Theory Model		Experimental Results of PMUT	
	Damping Ratio *ζ_t_*	Natural Frequency (Hz) *ω_t_*	Damping Ratio *ζ_e_*	Natural Frequency (Hz) *ω_e_*
1.5 mm	1.927 × 10^−3^	20,124	0.0086	23,927
1.25 mm	1.759 × 10^−3^	22,045	0.0103	24,161
1 mm	1.574 × 10^−3^	24,647	0.0099	24,262
0.75 mm	1.363 × 10^−3^	28,460	0.011	24,339
0.5 mm	1.113 × 10^−3^	34,856	0.023	24,493

**Table 5 sensors-22-07471-t005:** Results of adjusting factors of natural resonant frequency, damping ratio, effective mass, and viscous damper for varied volumes of Helmholtz resonant cavity.

	Lc = 1.5 mm	Lc = 1.25 mm	Lc = 1 mm	Lc = 0.75 mm	Lc = 0.5 mm
Adjusting factor of natural resonant frequency *α*	1.1890	1.0960	0.9844	0.8552	0.7027
Adjust factor of damping ratio *β*	4.4629	5.8556	6.2897	8.0704	20.665
Adjusting factor of effective mass *γ*	0.7074	0.8325	1.0320	1.3673	2.0252
Adjusting factor of viscous damper *δ*	3.7536	5.3428	6.3895	9.4369	29.408
gain factor *Kp*	0.1587	0.1961	0.2083	0.2632	0.4348

**Table 6 sensors-22-07471-t006:** Peak frequency and peak gain comparison of experimental data and calculated results based on the proposed modified Helmholtz resonator model.

Cavity Length (mm)	0.5	0.75	1	1.25	1.5
Peak frequency by modeling (Hz)	24,478	24,364	24,237	24,107	23,976
Experimental peak frequency (Hz)	24,480	24,336	24,262	24,158	23,925
Error (%)	0.00	0.12	0.10	0.21	0.21
Peak gain by modeling	9.46	11.76	11.32	9.80	8.32
Experimental peak gain	9.45	11.96	10.52	9.52	9.23
Error (%)	0.11	1.67	7.60	2.94	9.86

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
