# Peer review of "Piezoelectric Micromachined Ultrasonic Transducer-Integrated Helmholtz Resonator with Microliter-Sized Volume-Tunable Cavity"

_sensors, 2022, doi:10.3390/s22197471_

Round 1
Reviewer 1 Report
The authors present a PMUT integrated with a tunable Helmholtz resonator in order to improve the bandwidth and the sound pressure level of the transmitted acoustic wave. In the first part of the paper, the device is fabricated and the performance characteristics are experimentally measured. In the second part of the paper, the authors introduce a 1D equivalent circuit of the system in order to model its dynamic behavior. However, it is not very clear why this work is important or what application the authors are specifically targeting.
Furthermore, it appears that the first section on device characterization and fabrication has a major overlap with the contents in ref [21]. Many of the figures are an exact copy of those published in [21] and even the figure captions are the same. It is important the contents of the submitted manuscript be modified as this amounts to plagiarism. My other comments can be found below:
1. It would be very helpful to the reader if figure 1 can include a close up or a more detailed sketch of the actual PMUT structure. Right now, its very hard to understand the 4-beam structure.
2. Please include a figure with the fabrication steps as this is an important part of the description. Although the fab has been explained in detail in ref [22], it would be good to repeat it as it is relevant to this section.
3. pg 4, line 157 and 158: ‘the swept frequency range was set to 5V kHz….. at the swept 5 kHz’ – Can you please explain this more clearly as I am unable to understand this sentence.
4. Can you include the frequency response of the PMUT structure in free field (ie. In the absence of any cavity) in figure 4? This is very important as it provides a basis for the need for a Helmholtz resonator in the first place. The reader would definitely want to see the response and bandwidth of the PMUT when there is no enclosing structure present.
5. I am unconvinced with certain aspects of the equivalent electrical model of the integrated PMUT and Helmholtz resonator. Specifically, the radiation impedance of the fluid surrounding the PMUT top surface does not seem to be considered anywhere. From the equivalent model, the force Fa/P1 seems to act across the Re, Le and Ce which are lumped parameters representing the helmoltz cavity on the bottom side of the PMUT. Yet, Fa is calculated based on the observed pressure above the pmut membrane (which should probably be represented by some radiation term). Furthermore, on line 284 in page 10, it is assumed that h<
6. pg 12 line 331: it says that the measured output sound pressure is P(3), yet in figure 6, Fa is calculated based on the measured pressure. So what is P1, P2, P3 and Fa?
7. Can this model be verified experimentally under more conditions? The modified relations are only plotted against various parameters, but are not validated against experiment.
Author Response
Thank you very much for your time to review our manuscript and give us valuable comments and suggestions.

Reviewer 2 Report
The paper shows an interesting demonstration of a PMUT integrated Helmholtz resonator with a microliter-sized volume tunable cavity. Though the work is interesting, it needs a major revision with respect to the content and formatting. Kindly address all the comments given below.
Major comments
1. Authors mentioned that the Ti thickness is 4 micron. How did they handle this thin Ti layer? Did they mean 4 micron Ti was deposited on a substrate?
2. What is the thickness of the PZT film?
3. Please brief on the content of the precursor solution used for the growth of PZT film.
4. Characterization of the PZT film (such as orientation, SEM, piezo-properties etc.) will add value to the paper as PZT is one of the key elements of the device. Kindly include a few of them.
5. Specify the manufacturer of the microphone.
6. Ensure that all the units are given in the proper format. Eg. dB in figure 5 is given as ‘db’.
7. Page 3, line 100. “A 1 m-thick silver film was deposited …” is that 1 micron or something?
8. Comment on the mass manufacturing aspect of the device. Since many of the processes are kind of manual assembly, how do you plan to mass-manufacture the device?
9. Authors mentioned that they performed COMSOL analysis before fabricating the device, but the details are not included in the manuscript. Kindly include the details of FEM simulation performed using COMSOL.
10. Ensuring uniform format for all the graphs will make the manuscript appealing.
Author Response

(The authors gave the same response as above.)

Round 2
Reviewer 1 Report
Thanks for the response. The edits are satisfactory, however i still have concerns with a few details:
1. The authors have added some additional lines stating the benefits of this study in the introduction, but it would be good for readers if the authors could elaborate more on the potential applications of such a device and where it could be used.
2. According to line 208 on page 7, 'open cavity' has been defined as PMUT mounted on cavity but without the set screw. Although this cavity is open on one end, it still acts as a tube or a waveguide on the bottom side of the PMUT and would probably produce a different result as compared to a bottom support with less height. Is it possible to show via simulation that the height of the cavity 'L', might or might not make a difference to the frequency response in the absence of a screw?
3. The explanation regarding Fa, P1, P2 and P3 is a still a little vague and difficult to follow. In the response letter, the authors mention that Fa is the force causing the membrane to move. However, according to the equivalent circuit, the force acting across the membrane elements Cm, Rm and Lm is infact Fp-Fa. This means that the total force acting on the membrane due to the piezoelectric effect should be Fp and the force drop across the environmental impedance is Fa. This environmental impedance should account for both the top and the bottom fluid domains for the PMUT.
Also if P1 is the pressure measured 15 mm from the top surface, why is that being dropped across Re, Le and Ce in the equivalent circuit? The explantation does not match what is being represented on the equivalent circuit. I have doubts if a system such as this, which is exposed to very different impedance conditions on either side of the membrane can be adequately modelled with a simple two port network.
Author Response
Thank you very much for your valuable time to review our manuscript and give us comments and suggestions. Please check the attached file for our response.

Reviewer 2 Report
The authors have addressed all the concerns raised by this reviewer. Therefore, this manuscript may be accepted for publication.
Author Response
Thank you very much for your valuable time to review our paper.
Round 3
Reviewer 1 Report
The changes made to the manuscript are satisfactory, however, for the reasons stated in previous correspondence, I still believe the equivalent circuit model is incomplete in its current form. As the authors have mentioned, different impedance conditions on the top and bottom surface of the PMUT is indeed difficult to model, and I dont think the current two port representation is an accurate description of the dynamics- unless authors can show further evidence or by referencing a publication describing a similar system.
Author Response
Thank you for your valuable time to review our manuscript and give us comment and suggestion.
